# Study of Periodontal Bacteria in Diabetic Wistar Rats: Assessing the Anti-Inflammatory Effects of Carvacrol and Magnolol Hydrogels

**DOI:** 10.3390/biomedicines12071445

**Published:** 2024-06-28

**Authors:** Georgiana Ioana Potra Cicalău, Olivia Andreea Marcu, Timea Claudia Ghitea, Gabriela Ciavoi, Raluca Cristina Iurcov, Corina Beiusanu, Daniela Florina Trifan, Laura Grațiela Vicaș, Mariana Ganea

**Affiliations:** 1Department of Dental Medicine, Faculty of Medicine and Pharmacy, University of Oradea, 410068 Oradea, Romania; cicalau.georgiana@uoradea.ro (G.I.P.C.); gciavoi@uoradea.ro (G.C.); riurcov@uoradea.ro (R.C.I.); 2Department of Preclinics, Faculty of Medicine and Pharmacy, University of Oradea, 410068 Oradea, Romania; omarcu@uoradea.ro (O.A.M.); beiucorina@yahoo.com (C.B.); 3Department of Pharmacy, Faculty of Medicine and Pharmacy, University of Oradea, 410068 Oradea, Romania; laura.vicas@gmail.com (L.G.V.); mganea@uoradea.ro (M.G.); 4Department of Clinical Sciences, Faculty of Medicine and Pharmacy, University of Oradea, 410068 Oradea, Romania; trifan.daniela17@yahoo.com

**Keywords:** carvacrol, magnolol, periodontopathies, cytokines

## Abstract

Periodontal disease and diabetes often co-occur; both are characterized by chronic inflammation. This study aimed to investigate the anti-inflammatory effects of carvacrol and magnolol when incorporated into a periodontal hydrogel and topically applied to Wistar rats with diabetes-associated periodontal disease. Forty male albino Wistar rats were divided into four groups: PD (induced diabetes and periodontitis), PDC (induced diabetes and periodontitis treated with carvacrol), PDM (induced diabetes and periodontitis treated with magnolol), and PDCM (induced diabetes and periodontitis treated with both carvacrol and magnolol). Post treatment, gingival tissue samples were collected to measure levels of the pro-inflammatory cytokines IL-6 and TNF-α. The PDCM group exhibited significantly lower levels of interleukin-6 (IL-6) and tumor necrosis factor alpha (TNF-α) compared to the PD group. The combined application of a periodontal hydrogel containing carvacrol and magnolol may significantly reduce gingival inflammation in rats with diabetes-associated periodontal disease.

## 1. Introduction

The incidence of periodontopathies is closely associated with the presence of bacteria in gingival crevicular fluid [1]. These bacteria not only directly impact oral health by causing periodontal disease but also have systemic implications due to their role in chronic inflammation [2]. Specifically, pro-inflammatory cytokines such as Tumor Necrosis Factor-alpha (TNF-α) and Interleukin-6 (IL-6) have been linked to the development of chronic obesity, cardiometabolic diseases, and other systemic conditions [3]. These connections have become increasingly evident with the advent of modern medical technologies like polymerase chain reaction (PCR), which detects bacterial genomic fragments. Conditions associated with these bacteria include intestinal inflammation related to *Fusobacterium nucleatum* [4], colorectal cancer, prostate cancer, and dementia [5].

Analyzing the composition of gingival crevicular fluid has significantly advanced disease prevention. These analyses focus on bacterial DNA, particularly from species directly correlated with periodontal diseases and indirectly associated with elevated levels of inflammatory mediators. Bacteria such as *Fusobacterium nucleatum*, *Bacteroides forsythus*, and *Treponema denticola* have been identified as potential contributors to oncogenic processes, while others are associated with severe conditions like endocarditis, systemic infections, and cytokine-mediated inflammation [6,7,8].

Inflammation and the resulting cellular damage are closely tied to cytokine production and release, triggered by various inflammatory stimuli [9]. Maintaining a balance between pro-inflammatory and anti-inflammatory cytokines is crucial for modulating inflammation intensity. Pro-inflammatory cytokines, such as IL-6, produced by macrophages and epithelial cells, promote bone resorption and tissue damage. Conversely, anti-inflammatory cytokines like IL-10 and IL-4 act to mitigate these negative effects [10].

TNF-α, released by mononuclear phagocytes, induces the production of acute-phase proteins and plays a significant role in the inflammatory response. IL-6, known for its context-dependent pro- or anti-inflammatory effects, influences processes such as B lymphocyte differentiation and proliferation, monocyte-to-macrophage conversion, and osteoclast activation [11].

Several natural extracts have demonstrated the ability to improve immunological parameters and reduce pro-inflammatory cytokine expression when applied in vitro to immune cells or administered in vivo to animals or humans [12,13,14].

Carvacrol and magnolol are nutraceuticals with well-known contents, concentrations, and effects, as well as side effects. Therefore, the use of these nutraceuticals in clinical trials can provide greater accuracy in assessing their main effects. In the last 5 years alone, more than 138 articles on PUBMED have investigated the anti-inflammatory effect of carvacrol [15], and 165 studies have examined magnolol and its anti-inflammatory effect [16]. Chronic inflammation is the basis of many chronic and autoimmune diseases. As observed in numerous studies, natural extracts with anti-inflammatory effects are becoming increasingly important. Testing the anti-inflammatory effects of natural extracts, such as carvacrol and magnolol, and establishing their targeted areas of use are primary objectives of scientific research, and these are also among the objectives of this study. Notably, carvacrol and magnolol have shown potential to reduce inflammation in various pathologies [17].

Building on these findings, this study investigates the anti-inflammatory and antibacterial effects of a periodontal hydrogel containing carvacrol and magnolol through combined administration. The objective of this in vivo experiment is to assess the modulatory impact of carvacrol (found in plants such as Origanum vulgare) and magnolol (isolated from the bark of Magnolia officinalis) on two key gingival biomarkers, IL-6 and TNF-α, which are central to mediating inflammation in periodontal disease coupled with diabetes mellitus. Should the results prove promising, this approach may offer a new perspective on prevention for individuals with periodontal disease, potentially reducing local inflammation and lowering the risk of chronic diseases.

## 2. Materials and Methods

### 2.1. Experimental Design

A total of forty male Wistar albino rats, aged 8 weeks and weighing 220 ± 20 g, were housed in a controlled environment maintained at 21 ± 2 °C with 70 ± 4% humidity and a 12 h light/12 h dark cycle. Five rats were housed per cage. They were provided a standard pellet laboratory diet and had unlimited access to water. The rats were acclimatized for one week prior to the experiment.

The Wistar rats were randomly assigned to one of four groups:

PD—rats with experimentally induced diabetes mellitus and periodontitis (10 rats);

PDC—rats with experimentally induced diabetes mellitus and periodontitis treated with carvacrol (10 rats);

PDM—rats with experimentally induced diabetes mellitus and periodontitis treated with magnolol (10 rats);

PDCM—rats with experimentally induced diabetes mellitus and periodontitis treated with both carvacrol and magnolol (10 rats).

### 2.2. Micro-IDent Test

The Micro-IDent test involved DNA analysis of bacteria present in the gingival crevicular fluid. Gingival fluid was collected using dry paper points when the pocket depth exceeded 4 mm with bleeding on probing, even in cases of excellent oral hygiene. Each paper point was placed in the kit box, sealed, and stored at 2–8 °C until undergoing further processing.

DNA extraction from the paper points was performed using a bacterial DNA extraction kit (HAIN Lifescience, Nehren, Germany) according to the manufacturer’s instructions. The extracted DNA was then used for amplification in a multiplex PCR (Micro-IDent^®^plus, HAIN Lifescience GmbH) to identify specific bacterial species.

### 2.3. Chemicals/Extracts and Drugs Used

Carvacrol and magnolol extracts, used for preparing the periodontal hydrogels for topical application in periodontal disease treatment, were procured from Sigma-Aldrich^®^. A single dose of Streptozotocin, used to induce experimental diabetes mellitus, was also obtained from Sigma-Aldrich^®^ (St. Louis, MO, USA). Carbopol was sourced from S.C. VITAMAR IMPORT EXPORT SRL. IL-6 and TNF-α ELISA kits were purchased through the importer BIO ZYME SRL.

### 2.4. Periodontal Hydrogels

For topical application on the marginal gingiva and gingival sulcus, carvacrol and magnolol were used at minimum inhibitory concentrations documented in the international literature: 400 µg/mL for carvacrol and 25 µg/mL for magnolol (15). These active ingredients were incorporated into bioadhesive hydrogels formulated with 1% Carbopol (Sigma-Aldrich), according to results from a previous study [18,19].

### 2.5. Induction of Periodontal Disease

Periodontal disease was experimentally induced by placing a surgical ligature around the gingival sulcus of the mandibular left first molar, using 0.8 mm diameter stainless steel and chromium orthodontic wire (Figure 1). The ligatures were kept in place for three consecutive months to encourage bacterial biofilm accumulation, leading to the development of gingival inflammation and periodontal disease. Following this period, the ligatures were removed, and a comprehensive examination was performed to evaluate the extent of periodontal disease [20,21,22,23,24].

### 2.6. Induction of Diabetes Mellitus

Experimental diabetes mellitus was induced by administering a single intraperitoneal dose of streptozotocin. Blood glucose levels were monitored both before and after streptozotocin administration to confirm the successful induction of diabetes mellitus [20,25,26].

### 2.7. Treatment with the Periodontal Hydrogels

To address experimentally induced periodontal disease in the diabetic rat model, periodontal hydrogels containing carvacrol and magnolol were topically administered to the gingival sulcus and surrounding oral mucosa. These hydrogels were applied twice daily, in the morning and evening, over the course of one month. In the PDCM group, carvacrol hydrogel was applied in the morning, while magnolol hydrogel was applied in the evening [27,28].

### 2.8. Tissue Sample Collection and Analyses

Biopsy specimens were obtained from the marginal gingiva and the neighboring oral mucosa to assess inflammatory markers. The tissue samples were homogenized in phosphate-buffered saline, and the resulting homogenates were utilized for the quantification of IL-6 and TNF-α, which are indicative of tissue inflammation.

### 2.9. Determination of Inflammatory Biomarkers

The concentrations of IL-6 and TNF-α were assessed using the enzyme-linked immunosorbent assay (ELISA) technique. Spectrophotometric measurements, expressed in pg/mL, were employed to determine these concentrations. The tracked secretion levels indicated how much IL-6 and TNF-α was produced and released into the circulation by immune cells, providing a comprehensive analysis of cytokine secretion.

### 2.10. Statistical Analysis

Statistical analysis was performed using SPSS 20 Software (New York, NY, USA). The ANOVA test, followed by the Scheffe test for pairwise group comparisons, was employed to evaluate the effects of the hydrogel treatments on the rat groups. A significance level of *p* < 0.05 was established [29].

## 3. Results

### 3.1. Analysis of Periodontal Bacteria

*Fusobacterium nucleatum* was detected in all 50 specimens. *Actinobacillus actinomycetemcomitans, Bacteroides forsythus*, and *Treponema denticola* were found in 41 research subjects, while *fuso* and *Campylobacter rectus* were present in 33 animals. *Capnocytophaga* spp. and *Porphyromonas gingivalis* were observed in 26 animals, and *Eikenella corrodens* and *Prevotella intermedia* were observed in 16 individuals. *Eubacterium nodatum* was detected in only 16 of the rats. The exploratory analysis is presented in Table 1.

A graphical representation of these percentages is provided in Figure 2.

### 3.2. Analysis of IL-6 and TNF-α Marker Levels

A descriptive and comparative statistical analysis of the rat groups was conducted based on the levels of IL-6 and TNF-α markers, as observed in the gingival tissue (Figure 3). The inflammatory parameters (IL-6 and TNF-α) were normally distributed, with skewness and kurtosis values between +3.000 and −3.000.

#### 3.2.1. IL-6 Level Analysis

The highest IL-6 level was recorded in the PD group at 137.1253, with statistically significant differences compared to that of PDCM (*p* < 0.05). Differences were also noted but were statistically insignificant between PD and both PDC and PDM (*p* > 0.05), as well as between PDC and PDM, and between PDM and PDCM.

The difference in IL-6 values between PD and PDC was 3.17 with a standard deviation (SD) of 7.12. Between PD and PDM, the difference was 20.99 with an SD of 9.61. Lastly, the difference between PD and PDCM was 28.79 with an SD of 18.45. All differences were statistically significant (*p* < 0.01).

The Bonferroni multiple comparisons for IL-6 levels revealed significant mean differences among the groups. Specifically, the mean difference between PD and PDC was 3.17 (*p* < 0.01); between PD and PDM, it was 20.99 (*p* < 0.01); and between PD and PDCM, it was 28.79 (*p* < 0.01). Comparing PDC with the other groups, the mean difference was −3.17 (*p* < 0.01) with PD, 17.82 (*p* < 0.01) with PDM, and 25.62 (*p* < 0.01) with PDCM. For PDM, the mean difference was −20.99 (*p* < 0.01) with PD, −17.82 (*p* < 0.01) with PDC, and 7.80 (*p* < 0.01) with PDCM. Lastly, for PDCM, the mean difference was −28.79 (*p* < 0.01) with PD, −25.62 (*p* < 0.01) with PDC, and −7.80 (*p* < 0.01) with PDM. All comparisons were statistically significant with a 95% confidence interval.

#### 3.2.2. TNF-α Level Analysis

The highest TNF-α level was recorded in the PD group at 12.9371, with statistically significant differences compared to PDCM (*p* < 0.05). Differences were also noted but were statistically insignificant between PD and both PDC and PDM (*p* > 0.05), as well as between PDC and PDM, and between PDM and PDCM. The Bonferroni multiple comparisons for TNF-α levels revealed that no significant differences were observed in the obtained values between PD and PDC, PD and PDM, or PD and PDCM. A graphical representation of these findings is presented in Figure 4.

### 3.3. Scheffe Test Analysis

When conducting the ANOVA test for the variables IL-6 and TNF-α in gingival tissue among the PD, PDC, PDM, and PDCM groups, significant differences were observed. The ANOVA coefficient yielded F = 25.217 with *p* < 0.01 for IL-6, and F = 15.649 with *p* = 0.001 for TNF-α, both indicating statistical significance.

To identify the specific pairs of groups with significant differences, the Scheffe test was employed. Notably, significant variations in IL-6 levels (*p* < 0.05) and TNF-α levels (*p* < 0.05) were identified between the PD group and the PDCM group. The Scheffe test results for gingival IL-6 and TNF-α for each group are presented in Table 2.

## 4. Discussion

Our study demonstrated that the administration of periodontal hydrogels containing carvacrol and magnolol in Wistar rats with periodontal disease associated with diabetes decreases the levels of IL-6 and TNF-α in gingival tissue, favoring a reduction in inflammation [30]. The exclusive topical application of carvacrol or magnolol resulted in statistically nonsignificant results.

Carvacrol and magnolol have recently been found to have various positive effects, including anti-inflammatory, antioxidant, antibacterial, and antidiabetic effects [17].

Specialist studies indicate that *Campylobacter rectus* and *Porphyromonas gingivalis* are associated with an inflammatory process mediated by IL-6 and TNF-α. In our study, 66% of rats with induced diabetes were found to have *Campylobacter rectus*, and 52% had *Porphyromonas gingivalis*, which may explain the inflammatory process linked to both periodontopathies and diabetes [30].

In vitro studies have elucidated the anti-inflammatory effects of carvacrol, showcasing increased levels of IL-10 and its capability to diminish the production of inflammatory mediators such as IL-1β [31]. Furthermore, carvacrol has been shown to inhibit cyclooxygenase enzymes [32]. Other researchers have reported similar findings, indicating that carvacrol extract positively influences a reduction in IL-1β, IL-4, and IL-8, albeit without significant effects on IL-6 and TNF-α. These discrepancies were attributed to methodological differences [15,33].

Additionally, in vivo investigations have been conducted. In an inflammatory pain model, carvacrol demonstrated a reduction in rat paw edema and lowered levels of IL-1β and Prostaglandin E2 [31]. In vitro experiments, coupled with a carrageenan-induced pleurisy model, revealed that carvacrol exerted its anti-inflammatory effects by inhibiting cytokines and leukotrienes, ultimately inhibiting inflammatory edema and leukocyte migration [34].

In our study, the exclusive application of carvacrol (PDC group) led to a decrease in gingival IL-6 and TNF-α values compared to those of the PD group; however, this difference was statistically insignificant.

We will now compare our findings with those of other researchers. Tabibzadeh Dezfuli et al. (2017) observed that once-daily oral administration of carvacrol in animals with streptozotocin-induced diabetes reduced the blood levels of IL-1β, IL-6, and TNF-α [35].

Similarly, Wei Zhao et al. (2020) reported the potential of carvacrol in alleviating vascular inflammation in a type 2 diabetes animal model using male C57BL/KsJ db/db mice. Their results indicated that after six weeks of carvacrol administration by gavage, the serum levels of inflammatory mediators IL-1β, IL-6, IL-18, and TNF-α had decreased [36].

Furthermore, in a ligation-induced periodontitis experiment in rats, Kuo Po-Jan et al. (2017) found that gingival TNF-α and IL-6, along with IL-1β, inducible nitric oxide synthase expression, and Matrix metallopeptidase 9 (MMP-9), were modulated by intragastrically administered carvacrol [37].

In an effort to enhance the efficacy of carvacrol in treating periodontitis, Dao-run Hu et al. (2023) successfully modulated the Mitogen-activated protein kinase (MAPK) signaling pathway, TNF signaling pathway, and IL-17 signaling pathway by combining carvacrol with a nitrite reductase light-responsive nanodrug delivery system [38].

Our study aimed to investigate the potential therapeutic effects of magnolol, which has received less attention compared to carvacrol, in the context of periodontitis and diabetes [17]. The side effects can include an unpleasant phenolic taste [39], which reduces adherence to the treatment. Like any medicinal substance, carvacrol and magnolol can have adverse effects, such as irritation [40] or allergies.

In our literature review, we discovered that the intraperitoneal administration of magnolol has been demonstrated to significantly ameliorate the inflammatory response, exhibiting therapeutic promise in retinal angiogenesis by reducing inflammatory cytokines. Moreover, the oral administration of magnolol is believed to inhibit neutrophil migration into gingival tissue in rats [36]. Recently, Liu et al. (2021) showed that magnolol treatment reduces advanced glycation end product (AGE)-induced IL-6 production [35]. Consequently, supplementation with magnolol could potentially serve as a valuable approach to modulating inflammation in individuals with periodontal disease and diabetes mellitus [38].

In our study, the observed results deviated from our expectations. While a single administration of magnolol hydrogel (PDM group) exhibited a higher trend in IL-6 and TNF-α values compared to those in the carvacrol hydrogel (PDC) group, these differences did not reach statistical significance.

The potentially superior anti-inflammatory activity of magnolol may be attributed to its ability to upregulate inflammatory mediators IL-6 and TNF-α through an antioxidant stress mechanism. This is achieved by blocking the formation of advanced glycation end products (AGEs) at the gingival level. AGE accumulation in periodontal tissues is known to accelerate the progression of periodontal disease in individuals with diabetes. This mechanism may involve the upregulation of inflammatory mediators IL-6 and TNF-α. Furthermore, AGE formation induces the production of reactive oxygen species (ROS), exacerbating the vascular damage implicated in various diabetic complications [15,34,35,36,37,38].

We found limited literature addressing the investigation of magnolol, both in general and specifically in periodontal disease associated with diabetes. The studies we identified mainly focused on its effects in periodontal disease without a diabetes association. Moreover, the administration methods of magnolol differed across these studies, including intraperitoneal, oral, or oral gavage, without incorporation into a gel vehicle [17].

In light of the statistically insignificant results observed in our study, future research could explore the potential of increasing the dose or concentration of the active substance and prolonging the treatment duration to enhance the therapeutic effects.

Combining the administration of the two hydrogels in the PDCM group led to a significant decrease in IL-6 and TNF-α cytokines at the gingival level in periodontal disease associated with diabetes mellitus. Our findings suggest that the combination of these two extracts is particularly effective in reducing local inflammation in the gingival tissue of experimental animal subjects.

However, one limitation of our study is the statistically nonsignificant results, indicating the potential for higher doses or longer treatment durations to improve outcomes. Although our study demonstrated significant reductions in IL-6 and TNF-α cytokines with combined hydrogel administration, further research is warranted to elucidate the mechanisms underlying this synergistic effect, given the limited existing literature on the combined anti-inflammatory effects of carvacrol and magnolol. The biotransformations undergone by carvacrol, in comparison to magnolol, as well as the quality of the plants from which they are extracted, can change the anti-inflammatory effect over time. The quality conditions of the extractions and the stability of the natural products still require additional studies.

The enhanced anti-inflammatory efficacy of the carvacrol and magnolol combination may stem from its predominant modulation of distinct molecular pathways implicated in periodontitis associated with diabetes. Consequently, their simultaneous administration could potentially lead to a synergistic effect. Notably, we found no existing literature on in vitro or in vivo research assessing the combined anti-inflammatory effects of carvacrol and magnolol, underscoring the novelty of this study. Investigating the mechanisms underlying the enhanced therapeutic outcomes achieved with the combined use of carvacrol and magnolol represents a promising avenue for future research.

## 5. Conclusions

The reduction in IL-6 and TNF-α levels observed in gingival tissue following combined carvacrol and magnolol treatment implies a potential correlation between treatment efficacy and its influence on these specific inflammatory markers. It is conceivable that this combined therapy could have impacted the microbial composition in the gingival crevicular fluid and the oral microbiome, resulting in decreased IL-6 and TNF-α levels. This raises the possibility that carvacrol and magnolol, when used in combination, may possess antimicrobial properties that aid in regulating the presence of periodontal bacteria known to instigate inflammation. The observed reduction suggests a potential association between the treatment’s effectiveness and its influence on periodontal bacteria, underscoring the necessity for further research to elucidate this relationship. Additional investigations are warranted to explore this potential correlation comprehensively.

## Figures and Tables

**Figure 1 biomedicines-12-01445-f001:**
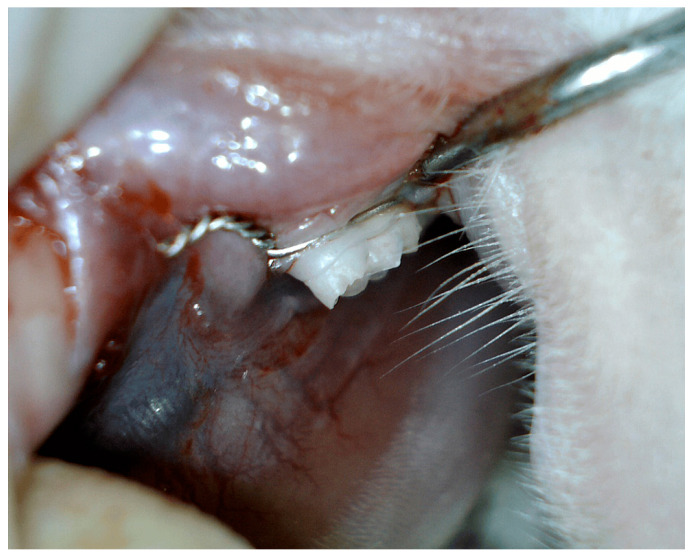
Induced periodontal disease in male Wistar albino rats.

**Figure 2 biomedicines-12-01445-f002:**
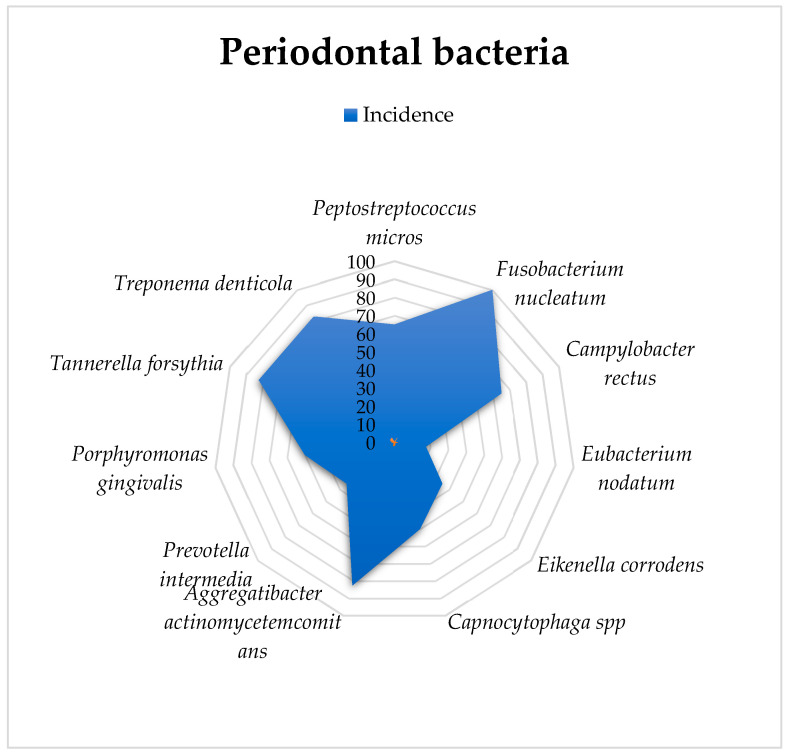
Graphical representation of the bacteriological analysis and incidence of infection in gingival crevicular fluid.

**Figure 3 biomedicines-12-01445-f003:**
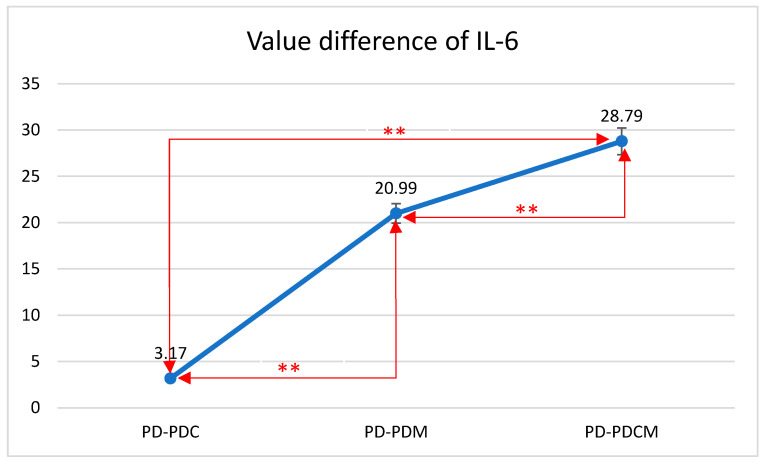
Mean differences in IL-6 levels in gingival tissue across all rat groups. IL-6—Interleukin 6, PD—rats with experimentally induced diabetes mellitus and periodontitis, PDC—rats with experimentally induced diabetes mellitus and periodontitis treated with carvacrol, PDM—rats with experimentally induced diabetes mellitus and periodontitis treated with magnolol, PDCM—rats with experimentally induced diabetes mellitus and periodontitis treated with both carvacrol and magnolol, **—Correlation is significant at the 0.01 level (2-tailed).

**Figure 4 biomedicines-12-01445-f004:**
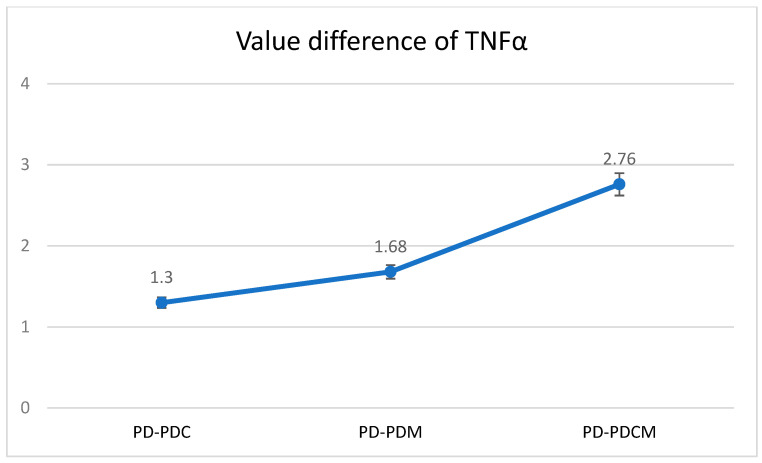
Mean difference of TNF-α levels in gingival tissue across all rat groups. TNF-α—Tumor Necrosis Factor-alpha, PD—rats with experimentally induced diabetes mellitus and periodontitis, PDC—rats with experimentally induced diabetes mellitus and periodontitis treated with carvacrol, PDM—rats with experimentally induced diabetes mellitus and periodontitis treated with magnolol, PDCM—rats with experimentally induced diabetes mellitus and periodontitis treated with both carvacrol and magnolol.

**Table 1 biomedicines-12-01445-t001:** The exploratory analysis of the presence of periodontal bacteria in gingival crevicular fluid.

Statistics	N		Mean	Std. Deviation	Skewness	Std. Error of Skewness	Kurtosis	Std. Error of Kurtosis	Range
	Valid	Missing							
*Peptostreptococcus micros*	124	0	0.6613	0.47519	−0.690	0.217	−1.549	0.431	1.00
*Fusobacterium nucleatum*	124	0	1.0000	0.00000	0.782	0.217	1107	0.431	0.00
*Campylobacter rectus*	124	0	0.6532	0.47787	−0.652	0.217	−1.601	0.431	1.00
*Eubacterium nodatum*	124	0	0.17	0.377	1.785	0.217	1.205	0.431	1.00
*Eikenella corrodens*	124	0	0.34	0.475	0.690	0.217	−1.549	0.431	1.00
*Capnocytophaga spp*	124	0	0.51	0.502	−0.033	0.217	−2.032	0.431	1.00
*Aggregatibacter actinomycetemcomitans*	124	0	0.83	0.377	−1.785	0.217	1.205	0.431	1.00
*Prevotella intermedia*	124	0	0.34	0.475	0.690	0.217	−1.549	0.431	1.00
*Porphyromonas gingivalis*	124	0	0.52	0.502	−0.065	0.217	−2.029	0.431	1.00
*Tannerella forsythia*	124	0	0.83	0.377	−1.785	0.217	1.205	0.431	1.00
*Treponema denticola*	124	0	0.83	0.377	−1.785	0.217	1.205	0.431	1.00

**Table 2 biomedicines-12-01445-t002:** Scheffe test results for gingival IL-6 and TNF-α in PD, PDC, PDM, and PDCM groups.

Variable	Groups		IL-6			TNF-α	
		Mean Difference	χ^2^	*p*	Mean Difference	χ^2^	*p*
PD	PDC	8.16356	0.116	0.978	1.29990	0.153	0.691
	PDM	25.97870	1.886	0.199	1.68368	0.515	0.416
	PDCM	33.77949 *	3.865	0.040 *	2.76074 *	4.128	0.033 *
PDC	PD	−8.16356	0.036	0.978	−1.29990	0.153	0.691
	PDM	17.81514	0.162	0.607	0.38378	0.028	0.998
	PDCM	25.61593	1.462	0.212	1.46084	0.261	0.575
PDM	PD	−25.97870	1.886	0.199	−1.68368	0.515	0.416
	PDC	−17.81514	0.162	0.607	−0.38378	0.028	0.998
	PDCM	7.80079	0.032	0.982	1.07706	0.107	0.831
PDCM	PD	−33.77949 *	3.865	0.040 *	−2.76074 *	4.128	0.033 *
	PDC	−25.61593	1.462	0.212	−1.46084	0.261	0.575
	PDM	−7.80079	0.032	0.982	−1.07706	0.107	0.831

*p*—statistically significant, IL-6—interleukin 6, TNF-α—tumor necrosis factor, χ^2^—Chi-Square coefficient, PD—rats with experimentally induced diabetes mellitus and periodontitis, PDC—rats with experimentally induced diabetes mellitus and periodontitis treated with carvacrol, PDM—rats with experimentally induced diabetes mellitus and periodontitis treated with magnolol, PDCM—rats with experimentally induced diabetes mellitus and periodontitis treated with both carvacrol and magnolol, *—correlation is significant at the 0.05 level.

## Data Availability

All the data processed in this article are part of the research for a doctoral thesis, which is being archived in the aesthetic medical office where the interventions were performed.

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
