# Peer review of "Study of Periodontal Bacteria in Diabetic Wistar Rats: Assessing the Anti-Inflammatory Effects of Carvacrol and Magnolol Hydrogels"

_biomedicines, 2024, doi:10.3390/biomedicines12071445_

Round 1

Reviewer 1 Report

Comments and Suggestions for Authors

Study of periodontal bacteria in diabetic wistaria rats: Assessing the anti-inflammatory effects of caracol and magnolia hydrogels.

1.     It is necessary to provide more details about Carvacrol and Magnolol. These extracts are plants, but what more?

2.The paper J. Apple. Oral Sci. 25(5) 2017. doi.org/10.1590/1678-7757-2016-0517 says about using a cotton ligature to induce periodontitis, but you use a wire ligature. Therefore, how do you evaluate pain? In the photo, the rat is blooding. How do you assess the suffering of the rat? It is essential to define this point because it may lead to animal misconduct. In addition, the time is very long. This paper says that in 15 days, bone loss was significant.

3.     How many rats were in a group/condition?

4.     Was a single intraperitoneal dose of streptozotocin sufficient to induce diabetes in all studies?

5.     Figure 2. To assess the presence of bacteria, you used a Micro-Ident test that analyzes the bacteria DNA and then performs an amplification in a multiplex PCR. Why don’t you present a graph pointing to the bacteria expression levels for the condition? It is correct to show the percentage of bacteria, but it is essential to consider the condition.

6.     Is the treatment with Carvacrol and Magnolol considered perpetuated in time?

7.     Figure 3 and 4. Is data significative? Please show it and put it in the manuscript.

8.     Line 206. Was your study in rats or mice?

9.     I suggest further discussing the possible complications and implications of using Carvacrol and Magnolol. While it is true that the study's limitations are described, more tools should be provided to improve the study.

Comments on the Quality of English Language

Minor editing of English language required.

Author Response

Reviewer 1

Firstly, we, the authors of the present manuscript wish to thank you for thoughtful commentary you have provided to improve the quality of the paper. We are very grateful for the time and effort you have devoted to this task. We have extensively revised my manuscript according to the recommendations. All changes in the text and the new figures that we have redesigned are highlighted. Please, see the correction highlighted in the manuscript.

Study of periodontal bacteria in diabetic wistaria rats: Assessing the anti-inflammatory effects of caracol and magnolia hydrogels.

  1. It is necessary to provide more details about Carvacrol and Magnolol. These extracts are plants, but what more?

Response: Thank you very much for observations. I have completed with the required information. (lines 60-69)

  1. The paper J. Apple. Oral Sci. 25(5) 2017. doi.org/10.1590/1678-7757-2016-0517 says about using a cotton ligature to induce periodontitis, but you use a wire ligature. Therefore, how do you evaluate pain? In the photo, the rat is blooding. How do you assess the suffering of the rat? It is essential to define this point because it may lead to animal misconduct. In addition, the time is very long. This paper says that in 15 days, bone loss was significant.

Response:  Thank you very much for observations. In this study, orthodontic wire specific to dentistry, approved by the animal research ethics committee, was used on rats to reduce animal stress. Gingival bleeding can be observed in a picture taken at the end of 7 days after the induction of periodontitis. The bone evolution of the rats was monitored for 14 days, but the ligature was fixed for only 7 days. The first 7 days were for acclimatization to reduce stress. I have added the required information. (lines 119-121)

  1. How many rats were in a group/condition?

Response: Thank you very much for observations. A total of forty male Wistar albino rats, divided in 4 equal groups, aged 8 weeks and weighing 220 ± 20 g, were housed in a controlled environment maintained at 21 ± 2 °C with 70% ± 4% humidity and a 12-hour light/12-hour dark cycle. Five rats were housed per cage. They were provided a standard pellet laboratory diet and had unlimited access to water. The rats were acclimatized for one week prior to the experiment. (lines 83-95)

  1. Was a single intraperitoneal dose of streptozotocin sufficient to induce diabetes in all studies?

Response: Thank you very much for observations. Each rat was between weighing 220 ± 20 g, each only received a single dose of streptozotocin. Line 108.

  1. Figure 2. To assess the presence of bacteria, you used a Micro-Ident test that analyzes the bacteria DNA and then performs an amplification in a multiplex PCR. Why don’t you present a graph pointing to the bacteria expression levels for the condition? It is correct to show the percentage of bacteria, but it is essential to consider the condition.

Response: Thank you very much for your comments. I corrected figure 2 and added the explanations.

  1. Is the treatment with Carvacrol and Magnolol considered perpetuated in time?

Response: Thank you very much for your comments. The biotransformations undergone by carvacrol, in comparison to magnolol, can change the anti-inflammatory effect over time, as well as the quality of the plants from which they are extracted. The quality conditions of the extractions, and the stability of the natural products, still require additional studies. (lines 298-302)

  1. Figure 3 and 4. Is data significative? Please show it and put it in the manuscript.

Response: Thank you very much for your comments. I have corrected figures 3 and 4 and indicate the significance.

  1. Line 206. Was your study in rats or mice?

Response: Thank you very much for your comments. I have corrected them.

  1. I suggest further discussing the possible complications and implications of using Carvacrol and Magnolol. While it is true that the study's limitations are described, more tools should be provided to improve the study.

Response: Thank you very much for observations. I have completed with the required information. (Lines 256-258)

Reviewer 2 Report

Comments and Suggestions for Authors

The manuscript entitled "Study of Periodontal Bacteria in Diabetic Wistar Rats: Assessing the Anti-inflammatory Effects of Carvacrol and Magnolol Hydrogels," by Georgiana Ioana Potra, Cicalău Andreea Olivia Marcu, Timea Claudia Ghitea, Gabriela Ciavoi, Raluca Cristina Urcov, Corina Beiusanu, Daniela Florina Trifan, Laura GraÈ›iela VicaÈ™, and Mariana Ganea, is a well-written and pleasant read. The manuscript aims to evaluate the anti-inflammatory effect of Carvacrol and Magnolol administered through a hydrogel (both together and separately) on animals induced with diabetes and periodontal disease. Although the work is clear and interesting, my concerns relate to the conclusions reached regarding the evaluated parameters. Indeed, the anti-inflammatory effect is assessed only by evaluating IL-6 and TNF-α, which are objectively insufficient to establish the effectiveness of the compounds.

The manuscript, in its current form, cannot be accepted for publication in Biomedicines as there are not enough experimental results to support the claims made in the discussion and conclusion. For example, the authors suggest that the effect of magnolol is also due to its activity against the formation of AGEs, but the study neither quantifies AGEs nor verifies the effect of the compound on animals with periodontal disease but not diabetes.

I believe the authors should enrich their interesting work with additional experiments to support their discussion and make the manuscript suitable for publication in Biomedicines.

Author Response

Reviewer 2

Firstly, we, the authors of the present manuscript wish to thank you for thoughtful commentary you have provided to improve the quality of the paper. We are very grateful for the time and effort you have devoted to this task. We have extensively revised my manuscript according to the recommendations. All changes in the text and the new figures that we have redesigned are highlighted. Please, see the correction highlighted in the manuscript.

The manuscript entitled "Study of Periodontal Bacteria in Diabetic Wistar Rats: Assessing the Anti-inflammatory Effects of Carvacrol and Magnolol Hydrogels," by Georgiana Ioana Potra, Cicalău Andreea Olivia Marcu, Timea Claudia Ghitea, Gabriela Ciavoi, Raluca Cristina Urcov, Corina Beiusanu, Daniela Florina Trifan, Laura GraÈ›iela VicaÈ™, and Mariana Ganea, is a wellwritten and pleasant read. The manuscript aims to evaluate the anti-inflammatory effect of Carvacrol and Magnolol administered through a hydrogel (both together and separately) on animals induced with diabetes and periodontal disease. Although the work is clear and interesting, my concerns relate to the conclusions reached regarding the evaluated parameters. Indeed, the anti-inflammatory effect is assessed only by evaluating IL-6 and TNFα, which are objectively insufficient to establish the effectiveness of the compounds.

The manuscript, in its current form, cannot be accepted for publication in Biomedicines as there are not enough experimental results to support the claims made in the discussion and conclusion. For example, the authors suggest that the effect of magnolol is also due to its activity against the formation of AGEs, but the study neither quantifies AGEs nor verifies the effect of the compound on animals with periodontal disease but not diabetes.

I believe the authors should enrich their interesting work with additional experiments to support their discussion and make the manuscript suitable for publication in Biomedicines.

Response: Thank you very much for your suggestion. I have revised the entire manuscript and marked the changes in red. Thank you very much for your observations. We have corrected the information related to the formation of AGEs and inflammation.

Round 2

Reviewer 1 Report

Comments and Suggestions for Authors

Thank you very much for take my comments but, I suggest that the data in figure 2 should be presented in a different way, because as the methodology is DNA extraction and then PCR amplification, it is necessary to evaluate the expression levels, not the percentage. To show the PCR data is by how many expression levels one group has increased compared to another and also to see if there is a difference between the groups. For this, at the statistical level, the exploratory analysis of the data, data contestation (if any), normal distribution of the data, followed by the statistical analysis itself and finally the post hoc analysis should be evaluated. 

With regard to the analysis of TNF-alpha and IL-6 levels, it is necessary to mention which levels are being assessed. Levels of secretion, expression, etc. And these data must be supported by the methodology, mentioning what was evaluated and how it was evaluated to obtain these results. And finally, the exploratory analysis of the data, data contestation (if any), normal distribution of the data, followed by the statistical analysis itself and finally the post hoc analysis should be evaluated. 

Comments on the Quality of English Language

Minor editing of English language required

Author Response

Reviewer 1

Thank you for your insightful comments and suggestions on our manuscript. Your feedback has been invaluable in enhancing the clarity and quality of our work. We greatly appreciate your careful consideration and the time you dedicated to reviewing our study.

Observation 1. Thank you very much for take my comments but, I suggest that the data in figure 2 should be presented in a different way, because as the methodology is DNA extraction and then PCR amplification, it is necessary to evaluate the expression levels, not the percentage.

Response 1. Thank you for the comment. We have changed the figure (line 168).

Observation 2. To show the PCR data is by how many expression levels one group has increased compared to another and also to see if there is a difference between the groups. For this, at the statistical level, the exploratory analysis of the data, data contestation (if any), normal distribution of the data, followed by the statistical analysis itself and finally the post hoc analysis should be evaluated. 

Response 2. Thank you for the observation. The exploratory analysis of the data was added, and figures 3 and 4 were modified and corrected (lines 181-200, 205-207, 209-215).

Observation 3. With regard to the analysis of TNF-alpha and IL-6 levels, it is necessary to mention which levels are being assessed. Levels of secretion, expression, etc. And these data must be supported by the methodology, mentioning what was evaluated and how it was evaluated to obtain these results.

Response 3. Thank you for the observation. Levels of secretion was analised. This specification was added in materials and methods section (lines 146-150).

Observation 4. And finally, the exploratory analysis of the data, data contestation (if any), normal distribution of the data, followed by the statistical analysis itself and finally the post hoc analysis should be evaluated. 

Response 4. Thank you for the observation. The exploratory analysis of the data was added (lines 174-175).

Reviewer 2 Report

Comments and Suggestions for Authors

The authors have significantly improved the manuscript; indeed, the discussion and conclusion are now consistent with the results. The presentation is better, and they have made the manuscript suitable for publication by better clarifying its meaning.

Author Response

Reviewer 2

The authors have significantly improved the manuscript; indeed, the discussion and conclusion are now consistent with the results. The presentation is better, and they have made the manuscript suitable for publication by better clarifying its meaning.

Response:

Dear Reviewer,

Thank you very much for accepting our manuscript for publication. We greatly appreciate your thorough review and valuable feedback, which have significantly improved the quality of our work. Your time and effort in evaluating our study are deeply appreciated. We are excited to see our research published and are grateful for your contribution to this achievement.

Sincerely,

The authors!